

# Morphology and molecular phylogeny of *Neolentinus* in northern China

Lei Yue[1,2,*], Yong-lan Tuo[1,2,*], Zheng-xiang Qi[1,2], Jia-jun Hu[1,3], Ya-jie Liu[1,2], Xue-fei Li[1,2], Ming-hao Liu[1,2], Bo Zhang[1,2], Shu-Yan Liu[1,2] and Yu Li[1,2,4]

[1] Engineering Research Centre of Edible and Medicinal Fungi, Ministry of Education, Jilin Agricultural University, Changchun City, Jilin Province, China
[2] College of Plant Protection, Jilin Agricultural University, Changchun City, Jilin Province, China
[3] School of Life Science, Zhejiang Normal University, Jinhua City, Zhejiang Province, China
[4] Mycological Valley Innovation Institute (Hefei), Hefei City, Anhui Province, China
[*] These authors contributed equally to this work.

## ABSTRACT

*Neolentinus* is a significant genus, belonging to Gloeophyllaceae, with important economic and ecological values, which are parasites on decaying wood of broad-leaf or coniferous trees, and will cause brown rot. However, the taxonomic study is lagging behind to other groups of macrofungi, especially in China. In view of this, we conducted morphological and molecular phylogenetic studies on this genus. We have discovered new types of cheilocystidia and with extremely long lamellae in *Neolentinus*, and, thus proposed it as a new species—*Neolentinus longifolius*. At the same time, we clarified the distribution of *Neolentinus cyathiformis* in China and provided a detailed description. Moreover, we also described two common species, viz. *Neolentinus lepideus* and *Neolentinus adhaerens*. All the species are described based on the Chinese collections. The key to the reported species of *Neolentinus* from China is provided. And the phylogeny of *Neolentinus* from China is reconstructed based on DNA sequences of multiple loci including the internal transcribed spacer (ITS) regions, the large subunit nuclear ribosomal RNA gene (nLSU), and the translation elongation factor 1-$\alpha$ gene (*tef-1$\alpha$*). In addition, full morphological descriptions, illustrations, color photographs, taxonomic notes, and all the available sequences of *Neolentinus* species are provided.

Corresponding authors
Bo Zhang, zhangbofungi@126.com
Shu-Yan Liu, liussyan@163.com

## INTRODUCTION

The genus *Neolentinus* Redhead & Ginns is overlooked in the family Gloeophyllaceae, while the taxonomic study in this genus started early. In 1799, Holmskjold described *Ramaria ceratoides* Holmsk. as a new species (*Holmskjold, 1799*), which was modified as *Agaricus polymorphus* var. *ceratoides* (Holmsk.) Pers., later, by *Persoon (1828)*. This is considered to be the same species as *Agaricus lepideus* Fr. described by *Fries (1815)*. After the establishment of the genus *Lentinus* Fr. (*Fries, 1825*), Fries transferred it into (*Fries, 1838*). Later, in the study of this genus, *Lentinus lepideus* was once proposed to be the type species (*Singer & Smith, 1946*; *Singer, 1986*), which contradicts the view of *Donk (1962)* and *Pegler (1983)*. It was not until *Redhead & Ginns (1985)* focused on the wood-rotting chemistry of the genus that the situation changed. They found that different species in *Lentinus* caused

two wood-rot types, white- or brown-rot, and, then it was used as a basis for establishing a new genus *Neolentinus* (*Redhead & Ginns, 1985*). The genus accommodates nine species in *Lentinus*, which almost coincides with the sect. *Squamosi* Fr., sect. *Pulverulenti* Fr., and sect. *Cirrhosi* Pegler in his monograph (*Pegler, 1983*) (except *Lentinus levis* (Berk. & M.A. Curtis) Murrill and *Lentinus sulcatus* Berk.), and takes *Neolentinus kauffmanii* (A. H. Sm.) Redhead & Ginns as the type species. At this point, the taxonomic status of the genus *Neolentinus* is becoming clear.

This genus so far consists of nine species, one variety, and two metamorphs. In 1998, Grgurinovic followed Pegler's opinion in his monograph (*Pegler, 1983*) and divided the genus into three groups, viz., sect. *Squamosi* (Fr.) Grgur., sect. *Cirrhosi* (Pegler) Grgur., and sect. *Pulverulenti* (Fr.) Grgur (*Grgurinovic, 1998*). Section *Squamosi* has a pileus surface, which is glabrescent or with appressed to recurved and fibrillose squama. Sometimes, there are partial veils and sclerotium in this group, however, the cystidia are missing. It is distributed in the north and south temperate (*Pegler, 1983*), and typified with *Neolentinus lepideus* (Fr.) Redhead & Ginns. Small basidiomata, terrestrial with a radicant stipe originating from a sclerotium, and cystidia absent are the characteristics of the sect. *Cirrhosi*. It is distributed in the paleotropical and Australia, the type species is *Neolentinus cirrhosis* (Fr.) Redhead & Ginns. And the sect. *Pulverulenti* is easier to distinguish, due to its convex or umbonate to applanate pileus and conspicuous elongated pleurocystidia that are lanceolate ventricose to cylindrical, frequently with refractive oleaginous contents, and occasionally with slightly thickened wall but not metuloidal. This group is widely distributed in Europe, Africa, South America, and North America, and is typified with *Neolentinus adhaerens* (Alb. & Schwein.) Redhead & Ginns. These three groups have the same hymenophoral trama structure and hyphal systems.

In the 1990s, molecular biology technology made a huge breakthrough. However, the molecular analysis of *Neolentinus* still lags behind than other genera. In 2011, *Garcia-Sandoval et al. (2011)* investigated the molecular phylogeny of the Gloeophyllales. The result shows that *Neolentinus* and *Heliocybe* Redhead & Ginns were clustered in the same clade, while *Neolentinus* is divided into five unequal subclades, corresponding to five different species. In 2016, *Zmitrovich & Kovalenko (2016)* conducted a molecular biology study of important lentinoid and polyporoid medicinal fungi. According to the phylogenetic results, species of *Neolentinus* clustered together, and are more distantly related to those of *Lentinus*. This also validates the scientific validity of the classification theory of Redhead and Ginns at the molecular level.

Because of the late start of research on macrofungi in China, it is also relatively lagged on *Neolentinus* research. In 1963, *Teng (1963)* described *L. lepideus* in *Fungi of China*. Later, in 1979, *Tai (1979)* recorded *Lentinus adhaerens* (Alb. & Schwein.) Fr. from Jilin and *Lentinus cyathiformis* (Schaeff.) Bres. from Yunnan. *Bi (1987)* compiled a species list of *Lentinus* of China in 1987, which includes *L. lepideus* (≡*N. lepideus*), *L. adhaerens* (≡*N. adhaerens*), and *L. cyathiformis* (≡*Neolentinus cyathiformis* (Schaeff.) Della Magg. & Trassin.). These are the only three species of the genus *Neolentinus* in the latest list (*Mao, 1996*; *Wang, Bau & Li, 2001*). However, there is no detailed description of *N. cyathiformis*, and based on the results of the specimens reviewed by *Li & Bau (2014)*, it is not certain whether this

species is distributed in China. At this point, the only clearly distributed species in China are *N. lepideus* and *N. adhaerens*.

In this article, 72 specimens of *Neolentinus* collected from northern China were studied based on morphology and molecular phylogeny. We have discovered and described one new species, one uncertain species, and two known species.

## MATERIALS & METHODS

### Sampling and morphological studies

The studied specimens were collected from northern China. Photos of fresh basidiomata were taken in the field. After measuring and describing the fresh macroscopic characteristics, the specimens were dried at 40–50 °C in the dryer, then they were stored in the Herbarium of Mycology of Jilin Agricultural University (HMJAU).

Macroscopic characteristics were based on the fresh basidiomata notes, and the colors were described according to *Küppers (2002)*. Referring to *Karunarathna et al. (2011)*, we make the following provisions according to the height (h) of the basidiomata: large size: h ≥ 10 cm; medium size: 5 cm ≤ h <10 cm; small size: h < 5 cm. Then microscopic characteristics were observed from the dried specimens using a Zeiss Axio lab. A1 light microscope. The dried specimens were rehydrated in 94% ethanol first, then mounted in 3% potassium hydroxide (KOH) to observe the color; sealed in 1% Congo red (0.1 g Congo red dissolved in 10mL distilled water) to measure the data; and observed in Melzer's reagent (1.5 g potassium iodide, 0.5 g crystalline iodine and 22 g chloral hydrate dissolved in 20 mL distilled water) to check if the spores are amyloid or dextrinoid (*Hu et al., 2022a*). As to each specimen, at least 20 values were measured separately from different basidiomata for each feature. The measurements are given as (a)b–c(d), the range of b–c contains a minimum of 90% of the measured values, and the extreme values (*i.e.,* a and d) are given in parentheses. The extent for basidiospores is given as length × width (L × W), Q values equal to the ratio of length and width of each basidiospore in the side view, "n" represents the number of measured basidiospores, "lm" represents the arithmetic mean of the length, "wm" represents the arithmetic mean of the width, and "q" represents the average Q value of all basidiospores ± standard deviation.

### DNA extraction, PCR amplification, and sequencing

The total DNA of the specimens was extracted using the new plant genomic DNA extraction kit from Jiangsu Kangwei Century Biotechnology Company Limited, following the instructions in an orderly manner. The amplification primers of the ITS regions were ITS1-F (CTT GGT CAT TTA GAG GAA GTA A) and ITS4-B (CAG GAG ACT TGT ACA CGG TCC AG) (*Gardes & Bruns, 1993*), the nLSU were LR0R (GTA CCC GCT GAA CTT AAG C) and LR5 (ATC CTG AGG GAA ACT TC) (*Cubeta et al., 1991*), and the *tef-1α* were 983F (GCY CCY GGH CAY CGT GAY TTY AT) and 2212R (CCR AAC RGC RAC RGT YYG TCT CAT) (*Rehner & Buckley, 2005*).

The amplification reactions were carried out using Kangwei Century 2 × Es Taq MasterMix (Dye) in a 25 μL system which is as follows: dd H2O 13.5 μL, 2 × Es Taq MasterMix (Dye) 8 μL, 10 mM primer 1 μL, 10 mM primer 1 μL, and DNA solution

1.5 µL. The PCR reactions procedure was as follows: for ITS, (1) 95 °C for 2 min to initial denaturation, (2) 35 cycles of denaturation for 40 s at 94 °C, annealing for 1 min at 50 °C, and extension for 2 min at 75 °C, (3) leave at 75 °C for 10 min (*Zmitrovich & Kovalenko, 2016*); for nLSU, (1) 95 °C for 3 min to initial denaturation, (2) 30 cycles of denaturation for 30 s at 94 °C, annealing for 45 s at 47 °C, and extension for 1min and 30 s at 72 °C, (3) leave at 72 °C for 10 min (*Hu et al., 2022b*); and for *tef-1α*, (1) 95 °C for 2 min to initial denaturation, (2) nine cycles of denaturation for 40 s at 95 °C, annealing for 40 s at 60 °C, and extension for 2 min at 70 °C, (3) then 36 cycles of denaturation for 45 s at 95 °C, annealing for 1 min and 30 s at 50 °C, and extension for 2 min at 70 °C, (4) leave at 70 °C for 10 min (*Zmitrovich & Kovalenko, 2016*). The PCR products were detected by 1.2% agarose gel electrophoresis, then entrusted Sangon Biotech (Shanghai) Company Limited to carry out purification and sequencing. Finally, the sequencing results were uploaded to GenBank (https://www.ncbi.nlm.nih.gov/genbank/), Table 1.

## Data analysis

By searching in GenBank, 12 ITS sequences, six nLSU sequences, and six *tef-1α* sequences of related taxa were downloaded. A total of 50 ITS sequences, 48 nLSU sequences, and 43 *tef-1α* sequences were obtained in this study based on 72 specimens of *Neolentinus*. All sequences used in this article are listed in Table 1. Each single-gene dataset was aligned in MAFFT 7 using the E-INS-i strategy (*Katoh & Standley, 2013*), and manually adjusted where necessary in BioEdit 7.0.9 (*Hall, 1999*). The datasets (ITS+nLSU, ITS+nLSU+*tef-1α*) were then concatenated using PhyloSuite 1.2.3 (*Zhang et al., 2020*; *Xiang et al., 2023*) for combined phylogenetic analyses. ModelFinder v2.2.0 (*Kalyaanamoorthy et al., 2017*) was used to select the best-fit model according to the AICc criterion. The best models are shown in Table 2. The datasets were analyzed separately using maximum likelihood (ML) and Bayesian inference (BI). The ML analyses were performed using IQ-Tree 1.6.12 (*Schmidt et al., 2014*), employing the optimal models as determined by Table 2. The tree topology was verified under both 1,000 bootstrap and 1,000 replicates of the SH-aLRT branch test. And the BI was performed using MrBayes 3.2.6 (*Ronquist et al., 2012*). The analysis employed a general time-reversible DNA substitution model and a gamma distribution to account for rate variation across the sites. Four Markov chains were executed for two runs, starting from random trees, for a total of 2,000,000 generations. The chains were terminated when the split deviation frequency value fell below 0.01. Tree samples were sampled every 1,000 generations. The first 25% of the sampled trees were discarded as burn-in, while the remaining trees were used to construct a 50% majority consensus tree and calculate the Bayesian posterior probabilities (BIPP). Then, FigTree v1.4.3 (*Andrew, 2016*) was used to visualize the resulting trees.

## Conservation status evaluation level and standards

According to The IUCN Red List of Threatened Species, Version 2021-3 (*IUCN, 2022*) and Red List of China's Biodiversity—Macrofungi (*Ministry of Ecology and Environment of the People's Republic of China, Chinese Academy of Sciences, 2018*), the conservation status of *Neolentinus* species in China is given.

**Table 1  Voucher/specimen ID, GenBank accession numbers, and origin of the specimens included in this study.** Sequences produced in this study are in bold.

| Taxon | Voucher/specimen ID | GenBank accession number | | | Origin | References |
|---|---|---|---|---|---|---|
| | | ITS | nLSU | *tef-1α* | | |
| *Neolentinus adhaerens* | UBC:F34010 | ON738525 | | | Canada | Unpublished |
| *N. adhaerens* | DAOM 214911 | HM536096 | HM536071 | HM536072 | Canada | *Garcia-Sandoval et al. (2011)* |
| **N. adhaerens** | **HMJAU6000** | **OR464178** | **OR464226** | **OR512975** | **China** | **This study** |
| *N. cyathiformis* | CTB 67-02 B | EF524038 | | | Germany | Unpublished |
| *N. cyathiformis* | CBS 244.39 | MH855996 | | | Netherlands | *Vu et al. (2019)* |
| *N. cyathiformis* | PV1109 | MG973063 | MG973062 | | Hungary | Unpublished |
| *N. cyathiformis* | LE3741 | KM411461 | KM411477 | KM411492 | Russia | *Zmitrovich & Kovalenko (2016)* |
| **N. cyathiformis** | **HMJAU67779** | **OR464176** | **OR464224** | **OR512974** | **China** | **This study** |
| *N. dactyloides* | CBS 258.49 | MH856514 | MH868044 | | Netherlands | *Vu et al. (2019)* |
| *N. kauffmanii* | iNAT-105732154 | OP020445 | | | USA | Unpublished |
| *N. kauffmanii* | DAOM 214904 | HM536097 | HM536073 | HM536118 | Canada | *Garcia-Sandoval et al. (2011)* |
| *N. kauffmanii* | CBS 315.50 | MH856643 | MH868153 | | Netherlands | *Vu et al. (2019)* |
| *N. lepideus* | BSE 2-2 | ON611827 | | | USA | Unpublished |
| *N. lepideus* | DAOM 208724 | HM536098 | HM536075 | HM536076 | Canada | *Garcia-Sandoval et al. (2011)* |
| *N. lepideus* | SFC20170810_14 | MT044412 | | | South Korea | Unpublished |
| *N. lepideus* | ZYJ0862 | OR142398 | | | China | Unpublished |
| *N. lepideus* | ASIS23411 | KP004924 | | | Republic of Korea | Unpublished |
| *N. lepideus* | ASIS25750 | KP004961 | | | Republic of Korea | Unpublished |
| *N. lepideus* | ASIS25837 | KP004964 | | | Republic of Korea | Unpublished |
| *N. lepideus* | LE253834 | KM411453 | KM411470 | KM411485 | Russia | *Zmitrovich & Kovalenko (2016)* |
| *N. lepideus* | DSM:3097 | GQ337914 | | | Germany | Unpublished |
| *N. lepideus* | BAM Ebw.20 | EF524039 | | | Germany | Unpublished |
| *N. lepideus* | KMCC22802 | OL721767 | | | South Korea | Unpublished |
| **N. lepideus** | **HMJAU67056** | **OR211597** | **OR211610** | **OR230703** | **China** | **This study** |
| **N. lepideus** | **HMJAU67780** | **OR464177** | **OR464225** | **OR512969** | **China** | **This study** |
| **N. lepideus** | **HMJAU67781** | **OR464172** | **OR464220** | **OR512970** | **China** | **This study** |
| **N. lepideus** | **HMJAU67789** | **OR464174** | **OR464222** | **OR512971** | **China** | **This study** |
| **N. lepideus** | **HMJAU67790** | **OR464175** | **OR464223** | **OR512972** | **China** | **This study** |
| **N. lepideus** | **HMJAU67791** | **OR464173** | **OR464221** | **OR512973** | **China** | **This study** |
| **N. longifolius** | **HMJAU67052** | **OR211593** | **OR211606** | **OR230702** | **China** | **This study** |
| **N. longifolius** | **HMJAU67054** | **OR211595** | **OR211607** | **OR230700** | **China** | **This study** |
| **N. longifolius** | **HMJAU67788** | **OR712443** | **OR712640** | **OR731274** | **China** | **This study** |
| *N. ponderosus* | MEXU 30242 | OP430550 | | | Mexico | Unpublished |
| *N. ponderosus* | MEXU 30247 | OP430555 | | | Mexico | Unpublished |
| *N. ponderosus* | MEXU 30248 | OP430556 | | | Mexico | Unpublished |
| *N. ponderosus* | MEXU 30250 | OP430558 | | | Mexico | Unpublished |
| *Veluticeps abietina* | K(M):191545 | MZ159507 | | | USA | Unpublished |
| *V. africana* | CBS 403.83 | MH861619 | | | Netherlands | *Vu et al. (2019)* |
| *V. ambigua* | He821 | JQ844472 | | | China | *He & Li (2013)* |
| *V. fasciculata* | He 2321 | KT750961 | KT750962 | | China | Unpublished |
| *V. fimbriata* | iNaturalist 17297393 | MZ318309 | | | USA | Unpublished |
| *V. microspora* | He670 | JQ844471 | | | China | *He & Li (2013)* |

**Table 2** The best models based on ModelFinder.

| | ITS | ITS+LSU | ITS+LSU+*tef-1 α* |
|---|---|---|---|
| ML | GTR+F+I+G4 | K3Pu+F+I+G4 | TN+F+I+G4 |
| BI | GTR+F+I+G4 | HKY+F+I+G4 | GTR+F+I+G4 |

## Nomenclature

The electronic version of this article in Portable Document Format (PDF) will represent a published work according to the International Code of Nomenclature for algae, fungi, and plants, and hence the new names contained in the electronic version are effectively published under that Code from the electronic edition alone. In addition, new names contained in this work have been submitted to Fungal Names from where they will be made available to the Global Names Index. The unique Fungal Names number can be resolved and the associated information viewed through any standard web browser by appending the Fungal Names number contained in this publication to the prefix "https://nmdc.cn/fungalnames/namesearch/toallfungalinfo?recordNumber=". The online version of this work is archived and available from the following digital repositories: PeerJ, PubMed Central SCIE, and CLOCKSS.

## RESULTS

### Phylogenetic analyses

The final dataset contains 78 gene sequences, and 33 of them were newly generated with 11 ITS sequences, 11 nLSU sequences, and 11 *tef-1α* sequences. A general phylogeny of *Neolentinus* based on the ITS sequences from the Maximum likelihood phylogenetic analysis is shown in Fig. 1, the general phylogeny of *Neolentinus* based on combined ITS and nLSU dataset from the maximum likelihood phylogenetic analysis is shown in Fig. 2, and the general phylogeny of *Neolentinus* based on combined ITS, nLSU, and *tef-1α* dataset from the Bayesian phylogenetic analysis is shown in Fig. 3.

Three well-supported clades were recovered in the ML and BI phylogenetics (Figs. 1–3) viz. clade A, clade B, and clade C. Among them, the clade C corresponds to sect. *Pulverulenti*, already recognized by *Garcia-Sandoval et al. (2011)*. However, clade A and clade B split sect. *Squamosi* in two independent clades. The clade A contains three species of the sect. *Squamosi* viz. *Neolentinus dactyloides* (Cleland) Redhead & Ginns, *N. lepideus*, and *Neolentinus ponderosus* (O.K. Mill.) Redhead & Ginns. It is worth mentioning that *N. lepideus* is a complex group. According to the phylogenetic analysis results (Figs. 1–3), the *N. lepideus* complex is divided into three branches. Of these, materials from North America are clustered into one branch with a portion of materials from East Asia, another portion of materials from East Asia are clustered into a separate branch, and materials from Europe are clustered into another branch with the rest of the materials from Asia. And in clade B, a new species, *Neolentinus longifolius* L. Yue, Y.L. Tuo, B. Zhang & Y. Li sp. nov., is clustered into a separate clade with high support. Furthermore, a material from China is clustered with the *N. cyathiformis* from Europe as a sister branch to *N. longifolius*.

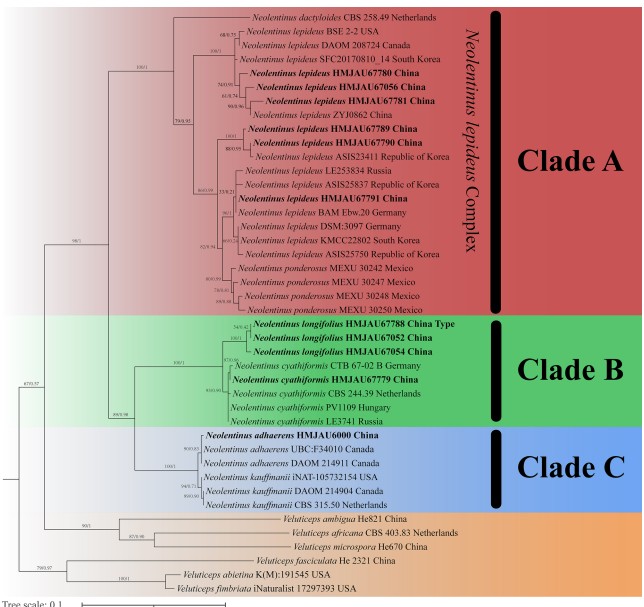

**Figure 1** **Fifty percent majority rule maximum likelihood analysis of *Neolentinus* based on ITS sequences, with *Veluticeps* species as outgroup taxa.** Support values of internal nodes respectively represent the maximum likelihood bootstrap (MLBP ≥ 70) and Bayesian posterior probability (BIPP ≥ 95%). The voucher or specimen ID and the country are marked after the species name and the sequence from the type specimen is also marked at the end.

## Taxonomy

*Neolentinus longifolius* **L. Yue, Y.L. Tuo, B. Zhang & Y. Li, sp. nov.**
Figs. 4A, 4B and 5

Fungal Names: FN571608

Etymology: The epithet "longifolius" refers to the extremely long lamellae.

Diagnosis: This species is distinguished from closed species by the lamellae that extend to the base of the stipe, the presence of apical branch and finger-like protrusion cheilocystidia on the stipe lamellae-edge, wider generative and skeletal hyphae, thinner pileipellis hyphae, and larger basidiospores.

Holotype: China. Jilin Province: Tonghua City, Liuhe County, Hani National Nature Reserve, 42.92°N, 126.19°E, 8 July 2023, Minghao Liu and Lei Yue, HMJAU67788.

Description: Basidiomata large. Pileus 10–20 cm in diameter, applanate, depressed and near flabellate, pale brown to yellowish brown ($N_{20}A_{60-80}M_{50}$), darker at the center, and lighter towards the margin, covered with squama and white ($N_{00}Y_{10}M_{00}$) flocculent hairs, stripe very light and radially parallel distributed, margin wavy and slightly involute. Lamellae deeply decurrent to the base of the stipe, subdistant, with 3 or 4 tiers of lamellulae, lamellae on the pileus stained white to grayish brown ($N_{10}Y_{20-40}M_{10}$), with serrate edge, while lamellae on the stipe stained white to pale yellow ($N_{00}Y_{10-20}M_{00-10}$), edge denticulate.

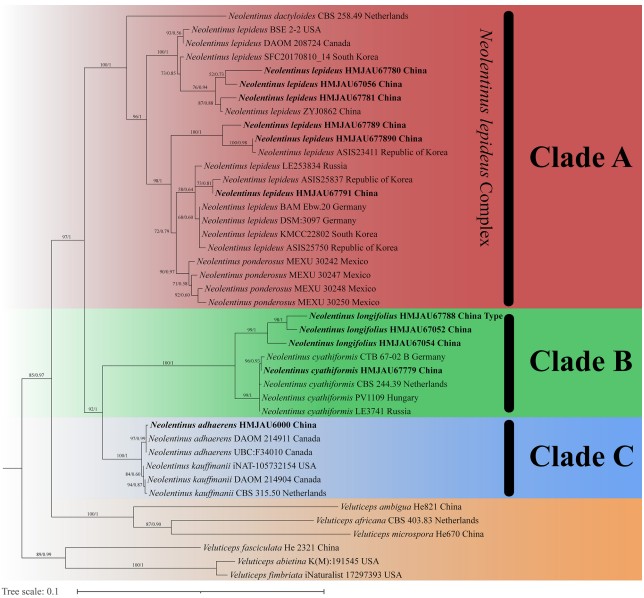

**Figure 2**  Fifty percent majority rule maximum likelihood analysis of *Neolentinus* based on ITS and nLSU sequences, with *Veluticeps* species as outgroup taxa. Support values of internal nodes respectively represent the maximum likelihood bootstrap (MLBP ≥ 70) and Bayesian posterior probability (BIPP ≥ 95%). The voucher or specimen ID and the country are marked after the species name and the sequence from the type specimen is also marked at the end.

Stipe 8–12 × 1.5–2.5 cm, inverted clavate, central or excentric, solid, coriaceous, with densely decurrent lamellae on the surface, upper part pale brown ($N_{60}A_{60}M_{50}$), basal enlargement, dark brown or blackish brown ($N_{90}M_{99}C_{99}$), surface reticulate and covered with numerous white ($N_{00}Y_{10}M_{00}$) flocculent hairs. Veil absent. Context thick, white ($N_{00}Y_{10}M_{00}$), coriaceous, fragrant, consisting of a dimitic hyphal system with skeletal hyphae.

Generative hyphae 3–8(10) μm diameter, sinuous, cylindrical, occasionally slightly inflated, hyaline, thin-walled or becoming distinctly thick-walled, frequently branched, lateral branches often form short and inflated vesicles, with prominent clamp-connections. Skeletal hyphae 3–6.5 μm diameter, sinuous cylindrical, hyaline, with a thickened wall and narrow lumen, unbranched or occasionally branched. Basidiospores (7.5)9–12(14) × (4)4.5–5.5(6) μm ($n = 40$, lm = 10.4 μm, wm = 4.73 μm, $Q = 1.5$–2.6, $q = 2.2$), ellipsoid to cylindrical, smooth, hyaline, thin-walled. Basidia (19)23–33 (37) × (5.5)6–8.5(9) μm, clavate, bearing 4 sterigmata. Lamella-edge sterile, with emergent, crowded cystidiiform hairs on the pileus lamellae-edge, 32–72 × 3–7 μm, sinuous cylindrical or subclavate, often constricted or nodulose, hyaline, thin-walked; cheilocystidia on the stipe lamella-edge (24)25–36(41) × (3)3.5–6 μm, with apical branch and finger-like protrusion, hyaline, thin-walled. Pleurocystidia absent. Hymenophoral trama subregular. Pileipellis a cutis, made up of thin-walled generative hyphae and a few thick-walled skeletal hyphae, slightly inflated, (2.5)3–6(7) μm wide, hyaline, smooth.

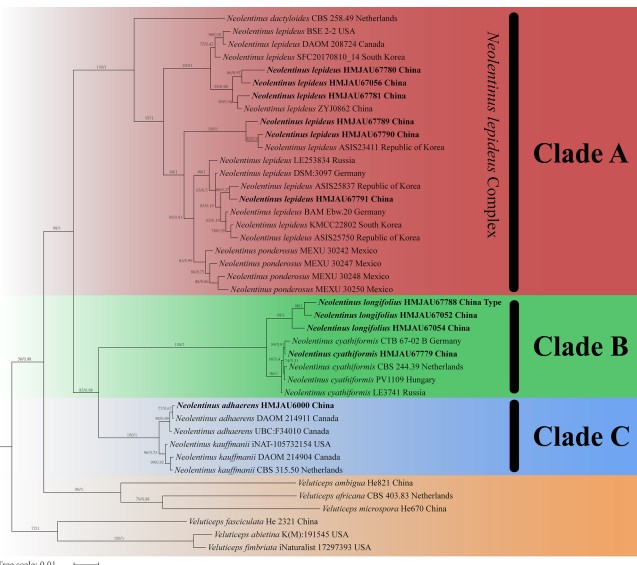

**Figure 3** **Fifty percent majority rule Bayesian phylogenetic analysis of *Neolentinus* based on ITS, nLSU, and *tef-1α* sequences, with *Veluticeps* species as outgroup taxa.** Support values of internal nodes respectively represent the maximum likelihood bootstrap (MLBP ≥ 70) and Bayesian posterior probability (BIPP ≥ 95%). The voucher or specimen ID and the country are marked after the species name and the sequence from the type specimen is also marked at the end.

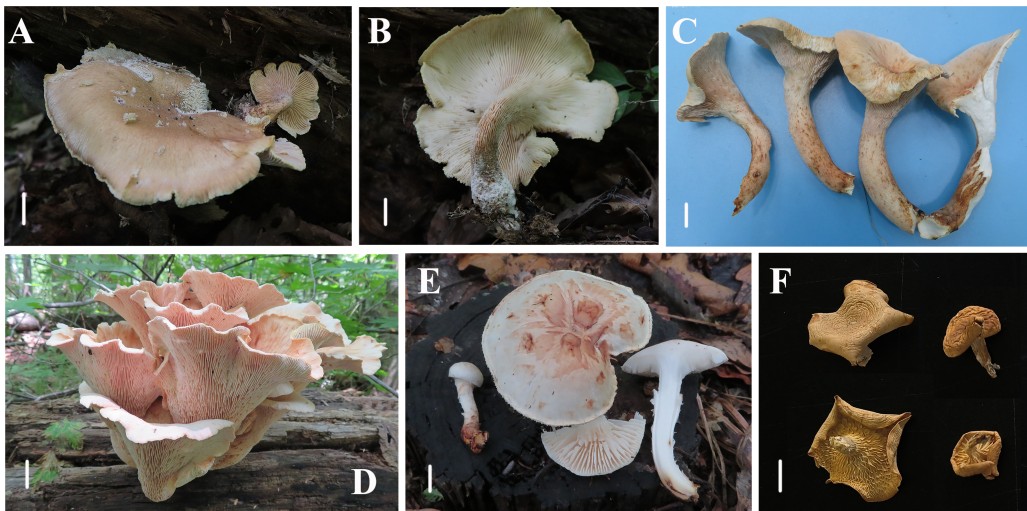

**Figure 4** **Basidiomata of *Neolentinus* species.** (A, B) *Neolentinus longifolius* (Holotype, HMJAU67788); (C, D) *Neolentinus cyathiformis* (HMJAU67779); (E) *Neolentinus lepideus* (HMJAU67781); (F) *Neolentinus adhaerens* (HMJAU6000). Scale bars = 1 cm.

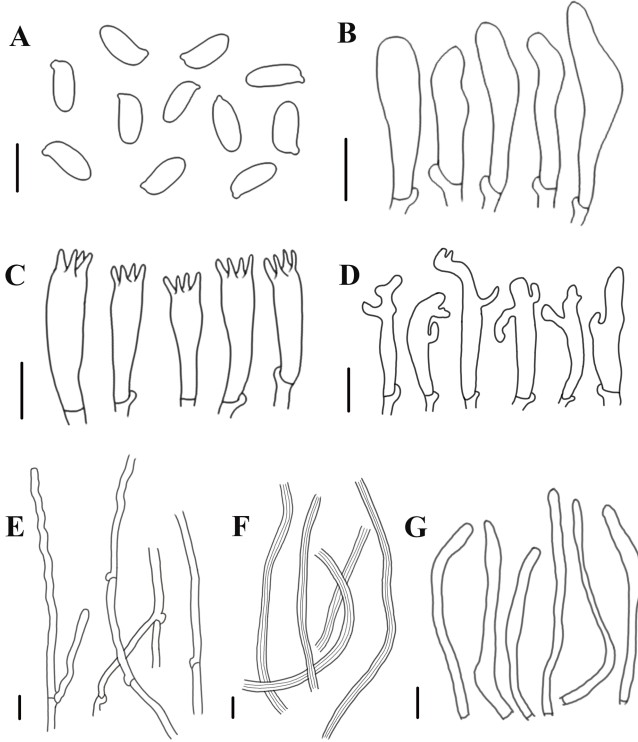

**Figure 5** **Morphological characteristics of *Neolentinus longifolius* (Holotype, HMJAU67788).** (A) Basidiospores, (B) basidioles, (C) basidia, (D) cheilocystidia on the stipe lamella-edge, (E) generative hyphae of context, (F) skeletal hyphae of context, (G) cystidiiform hairs on the pileus lamellae-edge. Scale bars: 10 μm.

Ecology: Scattered on the rotten woods in *Quercus mongolica* forest.

Distribution: Asia (China).

Conservation status: Not evaluated (NE).

Additional specimens examined: China. Jilin Province: Tonghua City, Ji'an County, Wunvfeng National Forest Park, 41.28°N, 126.12°E, 8 July 2021, Yonglan Tuo, HMJAU67052, Isotype (GenBank accession no.: ITS = OR211593, nLSU = OR211606, *tef-1α* = OR230702); Tonghua City, Ji'an County, Wunvfeng National Forest Park, Xianrentai Scenic Area, 41.26°N, 126.11°E, 16 July 2019, Yonglan Tuo, HMJAU60779; Tonghua City, Ji'an County, Wunvfeng National Forest Park, 41.27°N, 126.11°E, 3 July 2020, Yonglan Tuo, HMJAU67053; Tonghua City, Ji'an County, Wunvfeng National Forest Park, 41.27°N, 126.13°E, 5 July 2020, Jiajun Hu, HMJAU67054; Tonghua City, Ji'an County, Wunvfeng National Forest Park, 41.27°N, 126.11°E, 13 July 2019, Yonglan Tuo, HMJAU67777; Tonghua City, Ji'an County, Wunvfeng National Forest Park, 41.27°N, 126.13°E, 5 July 2020, Yonglan Tuo, HMJAU67778; Tonghua City, Ji'an County, Wunvfeng National Forest Park, 41.28°N, 126.12°E, 30 June 2023, Yonglan Tuo and Lei Yue, HMJAU67787.

Notes: This species is characterized by the extremely long lamellae that extend to the base of the stipe, and the presence of apical branch and finger-like protrusion cheilocystidia on the stipe lamellae-edge.

*Neolentinus longifolius* is close to *N. cyathiformis* in morphology, because of the decurrent lamellae and the similar pileus. Firstly, both of these species have deeply decurrent lamellae. However, the lamellae of *N. cyathiformis* only extend to the middle and upper part of their stipe, while the lamellae of *N. longifolius* extend to the base of the stipe. Secondly, their pileus surfaces all have hairs and squama. But the pileus of *N. cyathiformis* is hemispherical or convex to slightly depressed or cyathiform, the pileus of *N. longifolius* is applanate, depressed, and near flabellate. Thirdly, the stipe of *N. longifolius* is inverted clavate, the color of its upper part is pale brown while the expanded base is dark brown or blackish brown. Moreover, the expanded base surface is reticulate and covered with numerous white flocculent hairs. But the stipe of *N. cyathiformis* is subcylindrical or short and obconical, surface cream color and with reddish brown, furfuraceous squama or powdery punctatus. The base is blackish and tapering downwards. Fourthly, the lamellae on the stipe of *N. cyathiformis* are lacking cheilocystidia, while the lamellae on the stipe of *N. longifolius* have cheilocystidia, which have branch and finger-like protrusion at the apex. Fifthly, in terms of microscopic characteristics, *N. longifolius* has wider generative and skeletal hyphae, thinner pileipellis hyphae, and larger basidiospores than *N. cyathiformis*. Sixthly, they have different habitats. According to *Pegler (1983)*, *N. cyathiformis* grows on dead trunks, stumps, and branches of deciduous trees, especially *Salix*, *Populus*, *Tilia* and *Fagus*, but is rarely recorded in *Pinus*. While *N. longifolius* grows only on rotten woods in *Quercus mongolica*. Compared to *N. lepideus*, the latter has no cheilocystidia, the squama on the surface of the stipe is conspicuous, and sometimes there is a partial veil; but *N. longifolius* has cheilocystidia, inconspicuous squama on the stipe, and absent veil. Moreover, *N. leoideus* often has a whitish or yellowish-white pileus, while *N. longifolius* has a pale brown to yellowish brown pileus (*Pegler, 1983*; *Li & Bau, 2014*). In addition, compared to *N. ponderosus*, the stipe of *N. longifolius* is moderately thick, while the stipe of the latter is very thick. The pileus is different from the pinkish and cinnamon pileus of *N. ponderosus*. Furthermore, *N. ponderosus* has no velar tissue at all, while the pileus and stipe of *N. longifolius* both have white flocculent hairs (*Pegler, 1983*).

Because of the presence of the basidiomata that the lamellae extend to the base of the stipe in Europe, it does not seem convincing to consider it as a new species. However, this situation is rare in Europe, but it is widely present in Jilin Province, China. We have found the presence of this basidiomata on *Quercus mongolica* rotten woods in different years and different locations, which indicates that the characteristic is stably inherited. After microscopic observation, it was found that there were apical branch and finger-like protrusion cheilocystidia on the stipe lamellae-edge of these specimens. In summary, *N. longifolius* is robust as a new species.

*Neolentinus cyathiformis* (Schaeff.) Della Magg. & Trassin., Index Fungorum 171: 1 (2014)
Figs. 4C, 4D and 6

Description: Basidiomata medium to large. Pileus 5–6.5 cm in diameter, hemispherical or convex to slightly depressed or cyathiform, pale yellowish brown or grayish brown

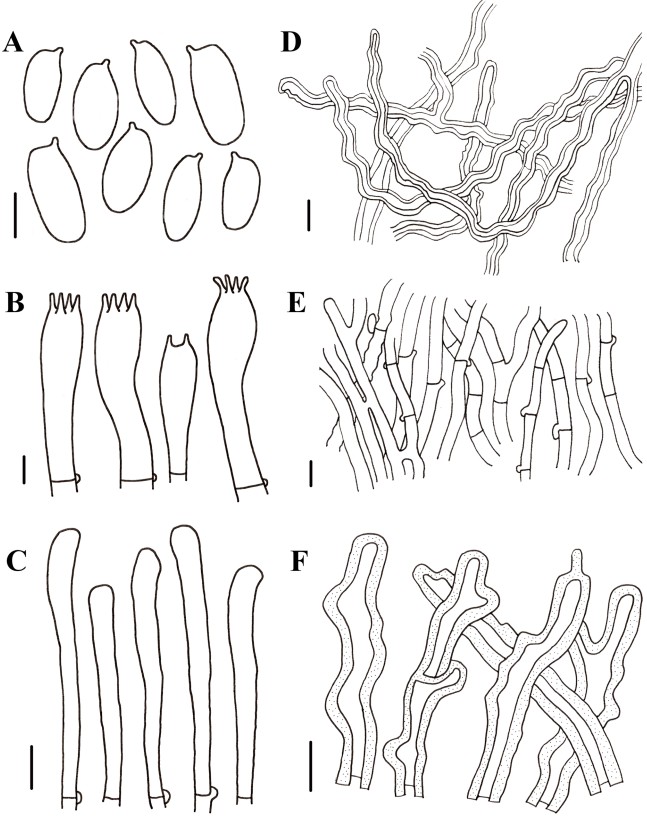

**Figure 6** **Morphological characteristics of *Neolentinus cyathiformis* (HMJAU67779).** (A) Basidiospores, (B) basidia, (C) cystidiiform hairs on lamellae-edge, (D) skeletal hyphae of context, (E) generative hyphae of context, (F) hairs on the pileus. Scale bars: (A, B) 5 μm, (C–F) 10 μm.

($N_{20}Y_{40-60}M_{10-20}$), often darker at the center, which is covered with densely white ($N_{00}Y_{10}M_{00}$) velvety short hairs, and yellowish brown ($N_{00}A_{60}M_{20}$) squama, edge slightly involute, sinuous. Lamellae deeply decurrent, and subdistant, with 3 or 4 tiers of lamellulae, cream color ($N_{00}Y_{20}M_{10}$), and edge denticulate. Stipe 6.2–8.1 cm, subcylindrical, central, solid, cream color ($N_{00}Y_{20}M_{10}$), with powdery punctatus on the surface. Veil none. Context thick, white ($N_{00}Y_{10}M_{00}$), coriaceous, consisting of a dimitic hyphal system with skeletal hyphae.

Generative hyphae 2.5–6.5 μm diameter, cylindrical, not inflated, hyaline, thin-walled or slightly thick, frequently branched, with prominent clamp connections. Skeletal hyphae 3–6 μm diameter, sinuous cylindrical, hyaline, with a thickened wall and continuous lumen, occasionally branched, finally tapering at the apex. Hairs on the pileus dense, 5–9 μm in diameter, with a thickened wall to 2.5 μm and elongated lumen. Basidiospores 8.5–11.5(12) ×4 –5(5.5) μm ($n = 40$, lm = 10.4 μm, wm = 4.6 μm, $Q = 1.9–3$, $q = 2.26$), ellipsoid to cylindrical, smooth, hyaline, thin-walled. Basidia (19)20–34(35.5) × (6)6.5–8(8.5) μm, clavate or elongated, bearing 4 or 2 sterigmata. Lamella-edge sterile, with emergent, crowded cystidiiform hairs, 3–6 μm diameter, elongated clavate, slightly inflated at the apex. Pleurocystidia absent. Hymenophoral trama subregular. Pileipellis a cutis, made up

of thin-walled generative hyphae and thick-walled skeletal hyphae, 3–10 μm diameter, hyaline, smooth.

Ecology: Scattered on the rotten woods in broad-leaved forests (*Populus, Tilia*).

Distribution: Asia (China), Europe.

Conservation status: Least concern (LC).

Specimens examined: China. Xinjiang Uygur autonomous region: Altay City, Kelan River Valley, 47.53°N, 88.00°E, 19 July 2022, Zhengxiang Qi, HMJAU67055; Jilin Province: Yanbian Korean Autonomous Prefecture, Antu County, Erdaobaihe Hydropower Station, 22 July 2018, Jiajun Hu, HMJAU67779.

Notes: In Pegler's monograph, the pileus of this species is hemispherical or convex to slightly depressed or cyathiform; the surface of the stipe is whitish to pale yellowish, darkening to tawny brown or blackish at the base, and covered with dense, small, reddish brown, furfuraceous squama; and generative hyphae of context are thin-walled, skeletal hyphae of context are unbranched (*Pegler, 1983*). In this study, the pileus has the same shape; the basidiospores and basidia have similar shapes and sizes. While the surface of the stipe is cream color with powdery punctatus. The generative hyphae and skeletal hyphae of context are slightly wider than Pegler's description, there are slightly thick walls in the generative hyphae, and skeletal hyphae are occasionally branched. Given previous reports on the distribution of *N. cyathiformis* in China (*Tai, 1979*; *Bi, 1987*; *Mao, 1996*; *Wang, Bau & Li, 2001*), the distribution of this species in China is questionable due to the inability to verify specimens and the lack of detailed descriptions (*Li & Bau, 2014*). In this study, we collected specimens of this species in China, confirming its distribution.

*Neolentinus lepideus* (Fr,) Redhead & Ginns, Trans. Mycol. Soc. Japan 26(3): 357 (1985)
Figs. 4E and 7

Description: Basidiomata small to medium. Pileus 1.6–8.7 cm in diameter, convex then applanate to depressed, surface whitish, yellowish-white, and pale yellowish brown ($N_{10-20}A_{50-60}M_{10-20}$), smooth, disrupting to form concolorous or dark brown ($N_{50}A_{99}M_{99}$) squama, largest towards the center, sometimes without squama, margin smooth, entire or sinuous. Lamellae sinuate decurrent, neither intervened nor anastomosing, subdistant, with 3 tiers of lamellulae, whitish or yellowish-white ($N_{10}A_{50-60}M_{10}$), edge dentate to lacerate. Stipe 2.2–6. 8 × 0.5–2.4 cm, cylindrical either tapering with a radicant base or sometimes with a bulbous base, central, solid, coriaceous, concolor with the pileus, blackish brown ($N_{90}M_{99}C_{99}$) below, surface with obvious squama. The veil forms a white ($N_{00}Y_{10}M_{00}$), peronate, fibrillose to membranous annulus to the stipe apex, usually ephemeral but rarely persistent. Context thick, white ($N_{00}Y_{10}M_{00}$), coriaceous, consisting of a dimitic hyphal system with skeletal hyphae.

Generative hyphae 4–9.5 μm diameter, sinuous cylindrical, not or occasionally slightly inflated, hyaline, thin-walled or becoming distinctly thick-walled, frequently branched, with prominent clamp connections. Skeletal hyphae 3.5–7 μm diameter, cylindrical, hyaline, with a thickened wall and continuous lumen, unbranched. Basidiospores (7.5)8.5–11(12.5) × 4–5.5 μm ($n = 40$, lm = 10.7 μm, wm = 4.86 μm, $Q = 1.88$–$2.38$, $q = 2.2$), ellipsoid

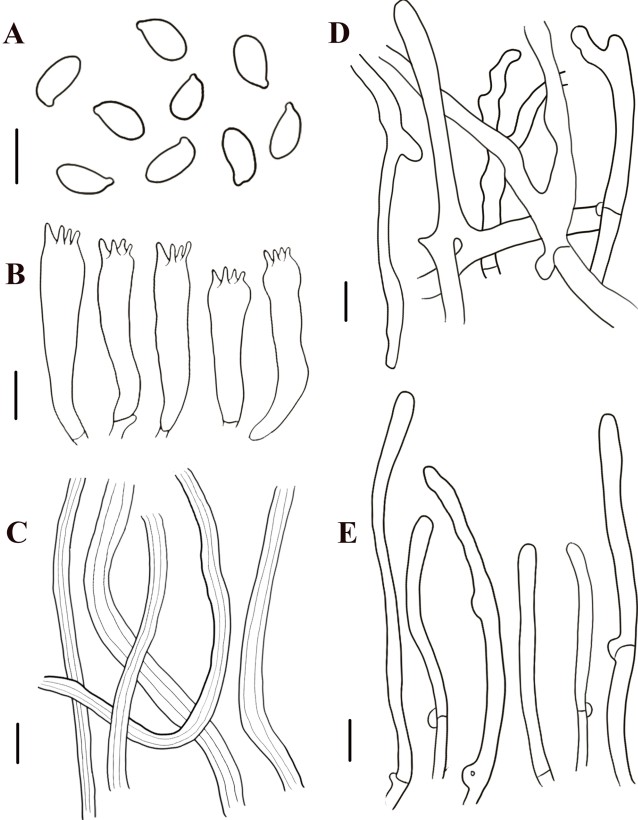

**Figure 7** **Morphological characteristics of *Neolentinus lepideus* (HMJAU67781).** (A) Basidiospores, (B) basidia, (C) skeletal hyphae of context, (D) generative hyphae of context, (E) cystidiiform hairs on lamellae-edge. Scale bars: 10 μm.

to cylindrical, smooth, hyaline, thin-walled. Basidia (30)31–40(41) × (7)7.5–8.5(9) μm, clavate, bearing four sterigmata. Lamella-edge sterile, with crowded cystidiiform hairs, 4.5–8.5 μm diameter, cylindrical or elongated clavate, hyaline, thin-walled. Pleurocystidia absent. Hymenophoral trama regular. Pileipellis an indefinite epicutis of radially repent generative hyphae, 2–9(15) μm diameter, with a hyaline thin-wall or a yellowish brown ($N_{20}A_{90}M_{50}$) thickened wall, unbranched, sometimes more or less slightly inflated.

Ecology: Scattered on the rotten woods and stumps of coniferous trees in coniferous or mixed coniferous and broad-leaved forests (*Abies, Picea, Pinus*).

Distribution: Asia, Europe, North America.

Conservation status: Least concern (LC).

Specimen examined: China. Xinjiang Uygur autonomous region: Hami City, Barkol Kazakh Autonomous County, East Tianshan Scenic Area, 43.31°N, 93.75°E, 7 July 2022, Zhengxiang Qi, HMJAU67056; Tarbagatay Prefecture, Bal Ruker Mount National Nature Reserve, 45.46°N, 82.45°E, 23 July 2022, Zhengxiang Qi, HMJAU67780; Jilin Province: Changchun City, Jingyuetan National Scenic Area, 8 June 2019, Jiajun Hu, HMJAU67781; Baishan City, Changbai Korean Autonomous County, 3 July 2019,

Jiajun Hu, HMJAU67782; Changchun City, Jingyuetan National Scenic Area, 29 May 2022, Bo Zhang, HMJAU67783; Yanbian Korean Autonomous Prefecture, Dunhua City, Hancongling Scenic Area, 16 July 2019, Gu Rao, HMJAU67789; Yanbian Korean Autonomous Prefecture, Dunhua City, Hancongling Scenic Area, 8 August 2020, Gu Rao, HMJAU67790; Changchun City, Changchun water culture ecological garden, 16 June 2022, Shengyue Zhang, HMJAU67791; Heilongjiang Province: Da Hinggan Ling Prefecture, Shuanghe National Nature Reserve, 22 June 2018, Dizhe Guo, HMJAU67784; Da Hinggan Ling Prefecture, Shuanghe National Nature Reserve, 8 August 2018, Dizhe Guo, HMJAU67785; Da Hinggan Ling Prefecture, Shuanghe National Nature Reserve, 23 June 2019, Dizhe Guo, HMJAU67786.

Notes: This species is the most widely known species of *Neolentinus* and was treated as the type species of *Lentinus* by Singer and Smith (*Singer & Smith, 1946*; *Singer, 1986*). Its distribution is widespread and varies considerably between regions and specimens, especially concerning characteristics such as basidiomata size, shape, pileus surface, and stipe surface. Some specimens growing in cold regions are lighter in color, while those in tropical and subtropical regions are generally more distinctly yellowish-brown. The specimens observed in this study had slightly smaller basidiospores and shorter but broader basidia than those described by *Pegler (1983)*, as well as broader cystidiiform hairs on lamella-edge, broader generative hyphae, and broader pileipellis hyphae. They all grow on decaying woods of coniferous trees.

*Neolentinus adhaerens* (Alb. & Schwein.) Redhead & Ginns, Trans. Mycol. Soc. Japan 26(3): 357 (1985)
Figs. 4F and 8

Description: Basidiomata small to medium. Pileus 2–8 cm in diameter, hemispherical or convex to applanate, slightly depressed or subcyathiform, surface whitish to ochraceous cream color ($N_{00-20}Y_{20-40}M_{60-70}$), pubescent, becoming viscid with a resinous, ambercolored secretion, finally almost laccate on drying and rich tawny brown ($N_{60}Y_{60}M_{30}$); margin initially inrolled, faintly plicate, undulate, eventually lobed. Lamellae short decurrent with a tooth forming a fine ridge down the stipe, occasionally anastomosing at the stipe apex, fairly broad, 4–6 mm wide, often splitting, distant, and with 4 tiers of lamellulae, pale yellow cream ($N_{10}Y_{20}M_{10}$), edge denticulate with resinous, brown ($N_{50}Y_{80}M_{30}$) droplets. Stipe 2–6 × 0.4–1 cm, cylindrical, radicant, central to slightly excentric, solid, surface concolor with the pileus, whitish ($N_{10}Y_{20-30}M_{10}$) pubescent becoming coated with a viscid, resinous secretion. Veil absent. Context thick, white to yellowish brown ($N_{00}Y_{20-40}M_{00-20}$), drying coriaceous and horny, consisting of a dimitic hyphal system with skeletal hyphae.

Generative hyphae 2–11(16) μm diameter, cylindrical, not inflated, hyaline, thin-walled or slightly thick-walled, frequently branched, often diverticulate, with prominent clamp connections. Skeletal hyphae 3–4 μm diameter, sinuous cylindrical, hyaline, thick-walled with a narrow lumen, unbranched. Basidiospores (4)4.5–7.5(8) × 2–4.5 μm ($n = 40$, lm = 5.82 μm, wm = 3.3 μm, $Q = 1.5$–2.33, $q = 1.76$), ellipsoid to cylindrical, smooth,

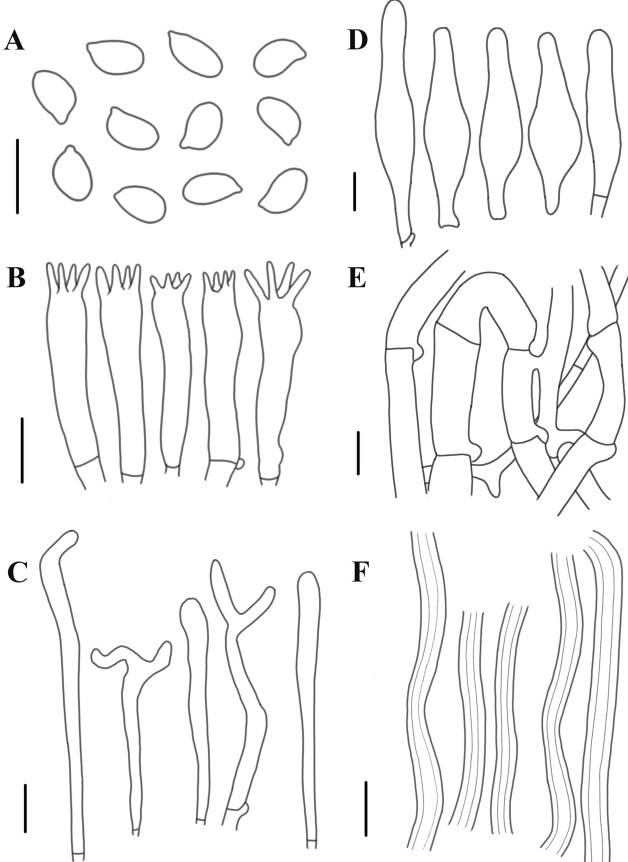

**Figure 8** **Morphological characteristics of *Neolentinus adhaerens* (HMJAU6000).** (A) Basidiospores, (B) basidia, (C) cystidiiform hairs on lamellae-edge, (D) pleurocystidia, (E) generative hyphae of context, (F) skeletal hyphae of context. Scale bars: 10 μm.

hyaline, thin-walled. Basidia 20–30 × 5–6.5 μm, clavate, bearing 4 sterigmata. Lamella-edge sterile, with crowded cystidiiform hairs, 3–5 μm diameter, cylindrical or subclavate, hyaline, thin-walled, often with a lateral branch at the apex. Pleurocystidia abundant, 33–71 × 7–15 μm, lanceolate with an inflated base and a tapering neck with an obtuse apex, often constricted, with a thin or slightly thickened wall, hyaline, with granular or refractive contents, sometimes coated with a resinous secretion. Hymenophoral trama regular, hyaline, of mostly generative hyphae. Pileipellis an indefinite epicutis of radially repent generative hyphae, 3–8(8.5) um diameter, with a hyaline thin-wall or a pale brown ($N_{20}A_{40}M_{20}$) thickened wall, unbranched.

Ecology: Scattered on the rotten woods in coniferous or mixed coniferous and broad-leaved forests (*Abies, Picea, Pinus*).

Distribution: Asia (China), Europe.

Conservation status: Least concern (LC).

Specimens examined: China. Xinjiang Uygur autonomous region: Fukang City, Changji Hui Autonomous Prefecture, Tianshan Heaven Pool Scenic Spot, 7 September 2007, Tolgor

Bau, HMJAU5668; Jilin Province: Baishan City, Fusong County, Lushuihe Town, 26 July 2006, Tolgor Bau, HMJAU6000.

Notes: The most distinctive feature that distinguishes this species from others is the reddish-brown resinous secretion on the surface of the pileus and stipe. The specimens observed in this study have smaller basidiospores, broader basidia, smaller pleurocystidia, and wider pileipellis hyphae than those described in Pegler's monograph (*Pegler, 1983*; *Li & Bau, 2014*).

### Key to the reported Chinese Species of *Neolentinus*

1. Lamellae decurrent to the base of the stipe, cheilocystidia on the stipe lamella-edge with apical branch and fingerlike protrusion.................................................................... *N. longifolius*
1. Lamellae decurrent to the apex of the stipe, cheilocystidia on the stipe lamella-edge absent ...................................................................................................2
2. Pileus and stipe surfaces, producing a viscid reddish-brown resinous secretion, with lanceolate with an inflated base and a tapering neck with an obtuse apex pleurocystidia.............................*N. adhaerens*
2. Pileus and stipe surfaces, lacking a viscid reddish-brown resinous secretion, pleurocystidia absent....................................................................................................3
3. Stipe surface often distinctly squama, partial veil developed often ephemeral.................. *N. lepideus*
3. Stipe surface with powdery punctatus, partial veil not developed.......................... *N. cyathiformis*

## DISCUSSION

### Morphological Characteristics of *Neolentinus*

Although the genus *Neolentinus* has only nine species now, they are widely distributed worldwide. Among them, *N. lepideus* is the most widely known species, usually distributed around the world, except the Antarctica and the Arctic (*Pegler, 1983*). However, the research on the genus in China is relatively fragmentary, with only two species officially reported, which are *N. lepideus* and *N. adhaerens* (*Li & Bau, 2014*). Systematic studies on this genus are uncommon in China.

In this study, 72 specimens collected from northern China were studied based on morphological and molecular phylogenetics, a total of one new species, one uncertain species, and two known species were identified and described, which increased the species diversity of *Neolentinus* in China. Among them, the new species was collected from the rotten wood in *Quercus mongolica* forest and produced brown rot. *N. cyathiformis* was collected from the broad-leaved forest. While *N. lepideus* and *N. adhaerens* were collected from rotten woods and stumps of coniferous trees in coniferous or coniferous and broad-leaved mixed forests. That is to say, different species have different habitats from one another. As saprotrophic fungi, all four of these species can degrade wood components, such as lignin, cellulose, and hemicellulose, indicating a crucial role in maintaining the stability of forest ecosystems (*Tuo et al., 2022*).

The new species identified in this study broadens the morphological characterization of the genus *Neolentinus*. In previous studies (*Pegler, 1983*; *Grgurinovic, 1998*; *Li & Bau, 2014*),

the lamellae-edge of species of this genus usually have only cystidiiform hairs, with only *Neolentinus papuanus* (Hongo) Redhead & Ginns has fusoid to short clavate cheilocystidia, while *N. longifolius* has cheilocystidia with branch and finger-like protrusion at the apex. In addition, the cheilocystidia of this species are found on the lamellae-edge of the stipe, while the lamellae-edge on its pileus bears numerous cystidiiform hairs. This means that there are some differences in the characteristics of the lamellae in different parts. However, previous studies have not focused on this feature in species that the lamellae extend onto the stipe.

## Phylogenetic Relationships of *Neolentinus*

In 2011, *Garcia-Sandoval et al. (2011)* carried out a molecular phylogenetic study of the Gloeophyllales and related taxa. Among them, the *N. kauffmanii* and *N. adhaerens* are divided into two branches, and all of them formed a sister clade with *N. lepideus*, with some genetic distance. And the same conclusion was reached in the present study. Based on these three analyses (Figs. 1–3), the genus *Neolentinus* is divided into three branches. Clade A is divided into five subclades containing three species, all of which belong to the sect. *Squamosi* (*Pegler, 1983*; *Grgurinovic, 1998*). The outermost species *N. dactyloides* is more distantly related to the other two species. However, *N. lepideus* is not a monophyletic species. On the basis of the results (Figs. 1–3), the *N. lepideus* complex is divided into three branches. Materials from North America and Europe are clustered in one branch each, while materials from Asia are distributed in all three branches. This seems to indicate a certain trend in the evolution of *N. lepideus*, in which Asia plays a crucial role. As for the different branches of *N. lepideus* in this study, only macroscopic subtle differences exist at present, and their microscopic differences still need to be further investigated. Clade B is divided into two subclades, *N. cyathiformis* and *N. longifolius*, which are more closely related and also belong to the sect. *Squamosi* (*Pegler, 1983*; *Grgurinovic, 1998*). Wherein, the material from China was clustered in a single branch with the *N. cyathiformis*, which also verified its distribution in China from a molecular point of view. Additionally, it is strongly shown that *N. longifolius* is a new species independent of *N. cyathiformis* because of the high degree of consistency in the results of these three analyses (Figs. 1–3). And clade C is divided into two subclades, namely *N. kauffmanii* and *N. adhaerens*, which are closely related to each other. These two species belong to the sect. *Pulverulenti* (*Pegler, 1983*; *Grgurinovic, 1998*).

This study increases the species diversity of *Neolentinus* around the world, especially from China, and provides new options for the exploitation of macrofungi resources. Morphological studies led to the same conclusions as the molecular systematics results. However, due to the lack of molecular data for some species in this genus, phylogenetic analysis of the entire genus cannot be conducted. This study found that the sect *Squamosi* is divided into two clades, so it is necessary to conduct further research on this group. Thus, in future work, it is essential to supplement the molecular data as much as possible and to research the species in sect. *Squamosi*, especially *N. lepideus* and *N. cyathiformis*, in more detail. This will make it possible to construct a phylogenetic tree of the whole genus,

and, thus clarify the phylogenetic relationships between groups and species, and reasonably explain the results found in this study.

## ACKNOWLEDGEMENTS

The authors thank Professor Tolgor Bau, Dr. Ao Ma, Mr. Dizhe Guo, and Dr. Gu Rao for the loan of the specimens studied; thank Dr. Yang Wang, Miss Xinya Yang, Miss Xinyue Gui, Miss Tongtong Tan, and Mr. Xin Wang for their help in the experiment; and thank the editors and reviewers for improving the manuscript.

### Funding

This research was funded by the Jilin Province Science and Technology Development Plan Project (No. 20230202112NC), the Wildlife and Plant Resources Investigation Project in Hani National Nature Reserve, Jilin Province (HX-2023003), the Research on the Creation of Excellent Edible Mushroom Resources and High Quality & Efficient Ecological Cultivation Technology in Jiangxi Province (20212BBF61002), and the Scientific and Technological Tackling Plan for the Key Fields of Xinjiang Production and Construction Corps (No. 2021AB004). The funders had no role in study design, data collection and analysis, decision to publish, or preparation of the manuscript.

### Grant Disclosures

The following grant information was disclosed by the authors:
The Jilin Province Science and Technology Development Plan Project: No. 20230202112NC.
The Wildlife and Plant Resources Investigation Project in Hani National Nature Reserve, Jilin Province: HX-2023003.
The Research on the Creation of Excellent Edible Mushroom Resources and High Quality & Efficient Ecological Cultivation Technology in Jiangxi Province: 20212BBF61002.
The Scientific and Technological Tackling Plan for the Key Fields of Xinjiang Production and Construction Corps: No. 2021AB004.

### Competing Interests

The authors declare there are no competing interests.

### Author Contributions

- Lei Yue conceived and designed the experiments, performed the experiments, analyzed the data, prepared figures and/or tables, and approved the final draft.
- Yong-lan Tuo conceived and designed the experiments, performed the experiments, analyzed the data, prepared figures and/or tables, and approved the final draft.
- Zheng-xiang Qi performed the experiments, analyzed the data, prepared figures and/or tables, and approved the final draft.

- Jia-jun Hu conceived and designed the experiments, performed the experiments, analyzed the data, prepared figures and/or tables, authored or reviewed drafts of the article, and approved the final draft.
- Ya-jie Liu performed the experiments, analyzed the data, prepared figures and/or tables, and approved the final draft.
- Xue-fei Li analyzed the data, prepared figures and/or tables, and approved the final draft.
- Ming-hao Liu analyzed the data, prepared figures and/or tables, and approved the final draft.
- Bo Zhang conceived and designed the experiments, authored or reviewed drafts of the article, and approved the final draft.
- Shu-Yan Liu conceived and designed the experiments, authored or reviewed drafts of the article, and approved the final draft.
- Yu Li conceived and designed the experiments, authored or reviewed drafts of the article, and approved the final draft.

## DNA Deposition

The following information was supplied regarding the deposition of DNA sequences:

*Neolentinus longifolius* L Yue, YL Tuo, B Zhang & Y Li, sp. nov. Fungal Names: FN571608.

The specimen is deposited in the Herbarium of Mycology of Jilin Agricultural University (HMJAU) which is one of the four major public mycological herbaria in China.

The specimen numbers are: HMJAU6000, HMJAU67052, HMJAU67053, HMJAU67054, HMJAU67056, HMJAU67779, HMJAU67780, HMJAU67781, HMJAU67789, HMJAU67790, HMJAU67791.

The sequences described here are available at GenBank: OR211593 to OR211595, OR211597, OR211606 to OR211608, OR211610, OR230700 to OR230703, OR464172 to OR464178, and OR464220 to OR464226.

## Data Availability

The tef-1$\alpha$ sequences of HMJAU6000, HMJAU67779, HMJAU67780, HMJAU67781, HMJAU67789, HMJAU67790 and HMJAU67791 are available in the Supplementary File.

## New Species Registration

The following information was supplied regarding the registration of a newly described species:

Neolentinus longifolius L. Yue, Y.L. Tuo, B. Zhang & Y. Li, sp. nov.

Fungal Names: FN571608

https://nmdc.cn/fungalnames/namesearch/toallfungalinfo?recordNumber=571608

## Supplemental Information

Supplemental information for this article can be found online at http://dx.doi.org/10.7717/peerj.16470#supplemental-information.

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
