# Peer review of "Morphology and molecular phylogeny of Neolentinus in northern China"

_PeerJ, doi:10.7717/peerj.16470_

## Round 0.1 · original submission · Minor Revisions

Please address the concerns of all reviewers and amend the manuscript accordingly.

Reviewer 1 ·

Basic reporting

No comment

Experimental design

No comment

Validity of the findings

No comment

Additional comments

The paper, titled "Morphology and Molecular Phylogeny of Neolentinus in Northern China," presents a taxonomic investigation of the Neolentinus genus within China. This research primarily focuses on conducting morphological and molecular phylogenetic analyses. The authors not only introduce a new species, Neolentinus longifolius but also furnish detailed descriptions of other species within the genus. Furthermore, they reconstruct the phylogeny of Neolentinus in China, employing DNA sequences. The paper offers a comprehensive package, including meticulous morphological descriptions, illustrations, color photographs, as well as taxonomic notes and available sequences of Neolentinus species.

This paper demonstrates several notable strengths:

1. Contribution to Taxonomic Knowledge: The paper significantly augments our taxonomic understanding of the Neolentinus genus in China, thereby addressing a substantial gap in the research pertaining to macrofungi in this geographical region.

2. Detailed Morphological Descriptions: The authors have thoughtfully provided intricate morphological descriptions, accompanied by illuminating illustrations and vivid color photographs. These resources prove indispensable for the precise identification and differentiation of distinct species.

3. Incorporation of Molecular Phylogenetic Analyses: By incorporating molecular phylogenetic analyses based on DNA sequences, the paper bolsters the taxonomic study's rigor. Additionally, it bestows valuable insights into the evolutionary relationships within the Neolentinus genus.

4. Logical Structure: The paper maintains a logical and well-organized structure throughout, facilitating readers in comprehending the research methodologies, findings, and conclusions seamlessly.

Nevertheless, there exist certain areas that warrant further attention from the authors:

1. Ecological Insights: It would be beneficial if the authors could furnish additional information regarding the ecological aspects of the studied species. Details such as habitat preferences and ecological roles would enhance the comprehensiveness of the paper.

2. Conservation Status: The paper currently omits information concerning the conservation status of the studied species. Including such data would be instrumental in understanding the conservation needs and priorities associated with these species.

·

Basic reporting

In this report, the author has described a new species of Neolentinus - N. longifolius characterized by long lamellae. Two main pieces of evidence were shown to support the claim of discovering a new species.
1) Morphological description of the fungi bodies
2) Genetic alignment of 2 loci, ITS and nLSU, with strong support of 100% ML/1 BI

Morphological studies were done from collected samples and preserved herbarium. Genetic alignment is done by comparing amplified loci of interest from the harvested samples to an online database of other sister taxa.

There are three branches of Neolentinus,
A) North America + some East Asia,
B) The rest of East Asia,
C) Europe.
The new species, N. longifolius, belongs to clade B.

This manuscript provides a detailed morphological and phylogenetic description of new species. The manuscript is well written with minimal grammatical errors and easy to follow. The figures are of good quality with minor mistakes. I think the manuscript should be considered for publication.

Experimental design

The samples were collected according to standard.
The detail of phylogenetics analysis is lacking. Please provide the following information
1. The setting of IQ-TREE, such as : the number of bootstraps, linkage calculation method, and method of likelihood testing.
2. The setting of MyBayes for Bayesian analysis, such as : how many replicates, how chains were run, and what was the stop criteria.

Validity of the findings

The findings are valid.

Additional comments

I have only minor comments below:
1. The new species' name should be mentioned in the manuscript title per journal guidelines.
2. Please check Figure 2, Clade B. The species identified as N. Lepideus has the same accession code as N.longifolius, according to Figure 1 and Table 1.
3. The position of N. lepideus on the tree in Figure 2 is incongruent with Figures 1 and 3.

Reviewer 3 ·

Basic reporting

Grammatic errors in line 22-23. It can be corrected to “At the same time, we clarified the distribution of Neolentinus cyathiformis in China and provided a detailed description. Moreover, we also described two common species, viz. ……”

Experimental design

No comment.

Validity of the findings

No comment.

Additional comments

This work did a systematic study on Neolentinus genus in northern China. This is an informative study that collected and examined 72 specimens from northern China using morphological and molecular phylogenetics, which lead to the identification of one new species, one uncertain species, and two known species. The paper is written in a professional and clear language. All necessary data, figures, methods are provided.

---

## Round 0.2 · accepted · Accept

Thank you for addressing issues pointed by all reviewers and for making the necessary amendments to the manuscript. I am glad to recommend your revised manuscript for publication.